# Higher Risk of Reoperation after Total Knee Arthroplasty in Young and Elderly Patients

**DOI:** 10.3390/ma16217012

**Published:** 2023-11-02

**Authors:** Albert T. Anastasio, Billy I. Kim, Niall H. Cochrane, Elshaday Belay, Michael P. Bolognesi, Grayson M. Talaski, Sean P. Ryan

**Affiliations:** 1Department of Orthopedic Surgery, Duke University, Durham, NC 27707, USA; albert.anastasio@duke.edu (A.T.A.); niall.cochrane@duke.edu (N.H.C.); elshaday.belay@duke.edu (E.B.); michael.bolognesi@duke.edu (M.P.B.); sean.p.ryan@duke.edu (S.P.R.); 2Department of Orthopedic Surgery, Hospital for Special Surgery, New York City, NY 10021, USA; kimbil@hss.edu; 3College of Engineering, University of Iowa, Iowa City, IA 52240, USA

**Keywords:** total knee arthroplasty, age, NSQIP, database, reoperation, complications

## Abstract

As outcomes and survivorship improve, total knee arthroplasty (TKA) has expanded into broader age groups. The purpose of this study is to analyze the impact of age on TKA outcomes using the National Surgical Quality Improvement Program (NSQIP) database from 2015 to 2020. Patients were categorized into young (40–49 years), middle (50–79 years), and elderly (80–89 years) groups. Findings reveal notable differences across age groups. The young cohort had the highest BMI, smoking incidence, and steroid use, while the elderly group exhibited a higher prevalence of comorbidities. Young patients experienced shorter hospital stays (*p* < 0.001) but longer operative times (*p* < 0.001), and outpatient surgery was most common in the middle age group. Multivariable regression demonstrated that the elderly group faced increased risks of pneumonia (*p* < 0.001), acute renal failure (*p* < 0.001), stroke (*p* < 0.001), cardiac arrest (*p* < 0.001), and transfusions (*p* < 0.001), while both young and elderly patients had higher 30-day reoperation risks (youngest cohort, 1.4% and elderly cohort 1.3% (*p* < 0.001)). In summary, elderly patients undergoing TKA are at the highest risk for medical complications, while young patients are more likely to undergo inpatient surgery, experience reoperations, and have longer operative times. This study underscores the importance of age-specific counseling for TKA patients and contributes valuable insights into the evolving landscape of knee replacement surgery.

## 1. Introduction

Total knee arthroplasty (TKA) involves metallic resurfacing of the proximal tibia and distal femur with placement of polyethylene articulating liner. TKA continues to rise in frequency across the United States (US). With the advent of outpatient TKA, further growth is predicted over the next decade [1,2,3]. As the incidence of TKA grows, mounting evidence suggests inferior outcomes in patients at the extremes of age.

Advancements in geriatric care and public health measures have resulted in extended “health expectancy” [4], as Americans are living longer. Thus, elective, quality-of-life procedures such as TKA are rising in incidence and are more commonly being performed in older individuals. Despite increased health expectancy, geriatric populations suffer from a higher rate of post-surgical complications across numerous surgical subspecialties [5,6,7,8]. Proposed mechanisms include vascular deterioration, decreased physiologic load to withstand the trauma of surgery, and immune system dysfunction [9,10]. Comorbidity burden is higher in older individuals and may account for the increased complication rate seen in these populations [11]. 

Additionally, TKA is becoming more frequent in younger age groups as indications expand for the utilization of this procedure [12]. Early data demonstrates conflicting results in TKA being performed in younger cohorts [13]; however, there is a dearth of literature documenting outcomes in age groups under the age of 50 years. 

While the relationship between complications and age has been previously explored, many of the previous studies are flawed in various ways, including single institution, non nationally representative sampling [14] and use of less recent data [15]. With recent advancements in perioperative care, an updated analysis is indicated to provide further insight in current trends of complication rates in populations at the extremes of age after TKA. Thus, the objective of this study was three-fold: to evaluate short-term complications, readmissions, and reoperations after TKA in young, middle, and elderly age group cohorts. The authors hypothesized that increased readmission rates, reoperation rates, and incidence of complications would be seen in the elderly age bracket and that the highest rates of outpatient TKA would be seen in the young age cohort. 

## 2. Methods

### 2.1. Patient Selection

Institutional Review Board Approval was obtained prior to conducting this study. The National Surgical Quality Improvement Program (NSQIP) database was queried for primary TKA (CPT code: 27447) between January 2015 and December 2020. Inclusion criteria included all patients underdoing a procedure with CPT code: 27447 without bilateral surgery or a concurrent case. Patients were excluded if they were under the age of 18 years, underwent a bilateral TKA, or underwent a concurrent procedure. The study population was then stratified by patient age at the time of primary TKA into three cohorts: young (40–49 years, n = 8895), middle (50–79 years; n = 286,988), and elderly (80–89 years; n = 125,780). 

### 2.2. Data Variables

All data variables were obtained from the NSQIP database. Patient demographics and baseline characteristics, including sex, race, ethnicity, inpatient vs. outpatient status, body mass index (BMI), current tobacco use (within one year of surgery), American Society of Anesthesiology physical status class (ASA), anesthesia type (general vs. other), surgical wound classification (clean, clean/contaminated, contaminated, vs. dirty), and financial status (independent vs. dependent) were collected. Medical comorbidities including diabetes, dyspnea, chronic obstructive pulmonary disorder (COPD), ascites, congestive heart failure, hypertension, renal failure, dialysis, disseminated cancer, open wound/wound infection, steroid use, weight loss, bleeding disorder, and receipt of pre-operative transfusion within 72 h of surgery were collected. Differences in patient demographics, baseline characteristics, comorbidities, and pre-operative laboratory values were compared across the groups using univariate analyses.

To asses differences in outcomes across age groups, the following outcome variables were collected: operative time, total hospital length of stay, discharge destination (home vs. not-home), as well as development of any of the following complications within 30 days of TKA: wound dehiscence, pneumonia, unplanned reintubation, pulmonary embolism (PE), acute renal failure, urinary tract infection (UTI), stroke/cerebrovascular accident (CVA), cardiac arrest requiring cardiopulmonary resuscitation (CPR), myocardial infarction (MI), intra- and post-operative requirement of transfusion, deep vein thrombosis (DVT) requiring therapy, sepsis, septic shock, reoperation and readmissions. The incidence of outcomes was compared across age groups using univariate analyses, and subsequently, multivariable regression analyses were performed to control for potentially confounding factors.

With regard to statistical analysis, continuous variables were represented as means and standard deviations, and categorical variables were reported as counts and percentages. The Shapiro–Wilk test was applied to demographic data to assess for normality of the included data. After confirmation of normality, univariate analyses were then performed using analysis of variance (ANOVA) for comparison of means and chi-square tests for comparison of counts across five groups. For post hoc analysis and to assess for confounding variables, multivariable regression was performed for each of the outcomes listed above with age group as the primary predictor and while controlling for pertinent baseline characteristics such as sex, race, ethnicity, inpatient vs. outpatient status, BMI, operative year, ASA class, and anesthesia type. Multivariable logistic regression was performed for binary outcomes (i.e., reoperation within 30 days) and linear regressions were employed for continuous outcomes (i.e., operative time). All statistical analyses were carried out in R v.3.6.1. Statistical significance was determined by *p*-value < 0.05.

## 3. Results

### 3.1. Patient Demographics and Baseline Characteristics

The prevalence of White/Caucasian patients increased by age group, comprising 78.3% of the young cohorts and 91.1% of the elderly cohort (*p* < 0.001) (Table 1). The prevalence of Asian patients also increased with age. However, the opposite pattern was present in the Black/African American (AA) cohort. Only 5.5% of the elderly cohort was Black/AA while 18.5% of the young cohort was Black/AA. Similarly, patients of Hispanic ethnicity had the highest representation (9.4%) in the young cohort and the lowest (5.7%) prevalence in the elderly cohort (*p* < 0.001).

The young cohort had the highest mean BMI of 36.6 kg/m^2^ (sd: 7.6) compared to 33.3 kg/m^2^ (sd: 6.6) and 29.2 kg/m^2^ (sd: 5.1) in the middle and elderly cohorts, respectively (*p* < 0.001). Rates of current tobacco use within the last year (*p* < 0.001) and steroid/immunosuppressant use for a chronic condition (*p* < 0.001) were also highest in the young cohort. Female gender patients demonstrated a similar bimodal pattern with increased rates of female patients in the young and elderly age groups (*p* < 0.001).

Overall, the elderly cohort had the highest ASA class (*p* < 0.001), less often received general anesthesia (*p* < 0.001), was less financially independent (*p* < 0.001), and had higher rates of comorbidities including dyspnea, COPD, hypertension, and bleeding disorders (*p* < 0.001 for all comorbidities). There were lower rates of outpatient TKA in the young and elderly compared to the middle age cohort (*p* < 0.001). 

### 3.2. Intra- and Peri-Operative Outcomes

Univariate analysis of perioperative outcomes revealed that the younger cohort had longer operative times (*p* < 0.001) (Table 2). Controlling for patient comorbidities, multivariable regression revealed that the young cohort had increased operative times whereas the elderly cohort had decreased times compared to the middle cohort (*p* < 0.001 for both, Table 3). The younger age groups were also associated with shorter lengths of hospital stay (*p* < 0.001) and lower rates of discharge to non-home facilities (*p* < 0.001).

### 3.3. Complications withing 30 Days of Surgery (30-Day Complications)

The elderly cohort had higher rates of nearly all 30-day complications with the exception of wound disruption and reoperation which were reported to be the highest in the young cohort (Table 2). Notably, rates of pneumonia, cardiac arrest, and DVT demonstrated a large increase in incidence from the middle to the elderly cohort, compared to a mild increase or lack of difference between the young and middle cohorts (Figure 1). Multivariable regression with the middle cohort as reference demonstrated that the elderly cohort had a significant increase in risk of pneumonia (OR: 2.1 [95%CI: 1.8–2.6]; *p* < 0.001), reintubation (OR: 2.1 [95%CI: 1.5–2.8]; *p* < 0.001), acute renal failure (OR: 2.4 [95% CI: 1.4–3.9]; *p* < 0.001), stroke/CVA (OR: 2.73 [95%CI: 1.9–3.8]; *p* < 0.001), cardiac arrest (OR: 2.9 [95%CI: 2.0–4.3]; *p* < 0.001), intra- or post-operative transfusions (OR: 1.8 [95%CI: 1.6–2.0]; *p* < 0.001), DVT (OR: 1.5 [95%CI: 1.3–1.7]; *p* < 0.001), and sepsis (OR: 1.4 [95%CI: 1.1–1.9]; *p* = 0.020) (Table 3).

All models included the following covariates: sex, race, ethnicity, ASA, anesthesia type (general vs. other), BMI, inpatient/outpatient surgery status, and operative year.

### 3.4. Readmissions and Reoperations

Reoperation rates within 30 days showed a bimodal distribution with the young cohort having the highest rate of 1.4% and the elderly cohort having the second highest rate of 1.3% (*p* < 0.001) (Figure 2). Multivariable regression revealed that the young (OR: 1.3 [95%CI: 1.0–1.5]; *p* = 0.024) and elderly (1.3 [95%CI: 1.2–1.5]; *p* < 0.001) cohorts had increased risk for 30-day reoperation compared to the middle cohort.

There was a stepwise increase in readmission rates between the young (2.7%), middle (2.8%), and elderly (5.3%) cohorts (*p* < 0.001). Controlling for other factors, regression analysis revealed an increase in 30-day readmission risk for the elderly (*p* < 0.001) but not for the young (*p* = 0.924) when compared to the middle cohort.

## 4. Discussion

As perioperative care and post-operative outcomes have improved, TKA has begun to be performed across a broader age distribution. There is a paucity of literature examining TKA across these expanded age groups. Previous investigations have inadequate sample size to allow for sufficient stratification by age cohorts and cannot be extrapolated to nationally generalizable data [16,17,18,19]. 

Easterlin et al. (48) most recently reviewed the NSQIP database to evaluate complications after TKA across age all groups. This analysis included a sample size of only 8950 patients from 2005 to 2009. Over the past decade, outpatient TKA has grown in feasibility across various orthopedic practice settings and in popularity amongst patients [20,21,22]. Thus, the present study aimed to update the literature and define complication rates in age-stratified cohorts while controlling for comorbid conditions. 

The current study indicates a higher percentage of patients in the oldest age group (80 to 89 years, 91.1%) are described as White/Caucasian. The reverse pattern was noted in the Black/AA cohort, comprising only 5.5% of the oldest age group. This racial disparity in TKA utilization is concerning, and is consistent with the prior literature in total joint arthroplasty which indicates that across all age groups, Black/AA cohorts are less likely to receive TKA than White/Caucasian cohorts [23]. 

The results of this study indicate that the middle cohort has the highest prevalence of outpatient billed TKA. Previous data has indicated that outpatient TKA may be poorly suited for patients over the age of 70 [24]; however, positive results have been reported in younger patients receiving outpatient TKA [25]. Interestingly, the results of the current study demonstrated the young cohort experienced the longest operative times, being on average 13 min longer than the oldest age cohort. These findings may be related to patient-specific factors which have led to a need for TKA at a younger age, such as post-traumatic arthritis or obesity. Significant joint deformity may contribute to longer operative times and obesity may contribute to difficulty with postoperative mobilization, and thus failure of outpatient TKA. Future research should aim to expand upon this finding and evaluate this seemingly paradoxical phenomenon where young age groups exhibit higher rates of inpatient TKA. 

The present study demonstrated increased post-operative complications with increasing age groups. Specifically, the largest increase in complication rates was noted between the middle age group and the elderly age group. These results are similar to those reported by Easterlin et al., where complication rates appeared to increase over the age of 70, with the sharpest increase in patients over the age of 80. Results from this study differ slightly from Higuera et al., who reported more step wise increase in complication rates [18].

Early reoperation rates also demonstrated a bimodal distribution, with the young age group and the elderly age group exhibiting the highest rates. This finding contradicts the author’s hypothesis, given prior research demonstrating higher rates of reoperation in advanced age cohorts. 

### Limitations

The NSQIP database is considered one of the most rigorously reviewed and accurate nationally representative databases available for query [26]. However, it is not without limitations, including those inherent to national outcomes databases [27]. The accuracy of the data is impacted by documentation practices and coding trends which may vary across hospitals. Specific to the NSQIP, data may be limited by exclusion of certain hospitals, particularly community-based hospitals where complication rates may differ substantially from larger university or large-system-affiliated hospitals. Additionally, NSQIP does not allow for inclusion of various data points specific to total joint arthroplasty. Finally, the database is limited to complications that occur within 30 days of discharge. Thus, the authors were unable to evaluate rates of chronic prosthetic joint infection and other delayed onset complications which cause substantial morbidity for TKA patients. 

## 5. Conclusions

In conclusion, the lowest rates of outpatient TKA occurred in the oldest age group, followed by the youngest age group. A general increase in complication rates was noted in the oldest age group compared to the middle and youngest age groups. Readmission rates were highest in the oldest age group, followed by the youngest age group. While high complication rates in older cohorts have been well documented, future research should aim to examine specific causality for higher readmission rates and lower rates of outpatient TKA in the youngest age group relative to middle age groups. 

## Figures and Tables

**Figure 1 materials-16-07012-f001:**
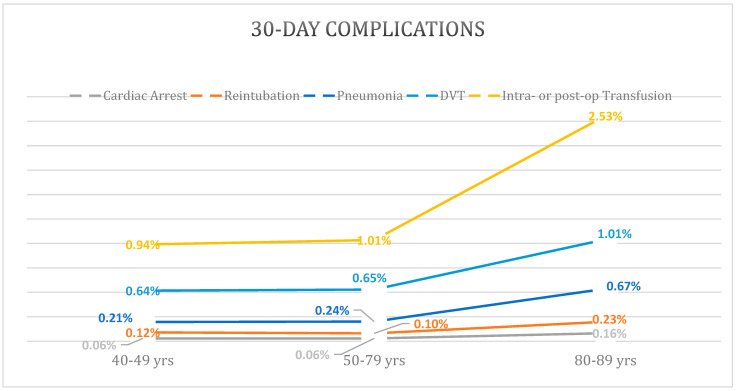
Increased Rates of 30-Day Complications in Elderly Age Group.

**Figure 2 materials-16-07012-f002:**
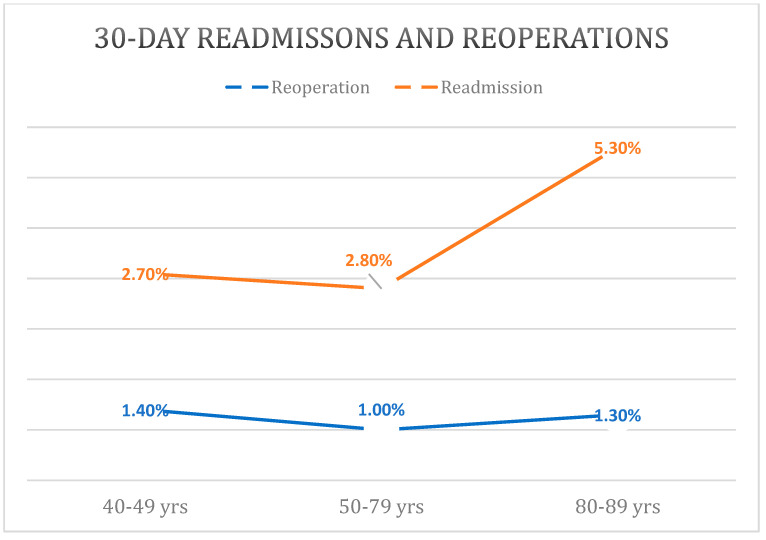
Higher Rates of 30-Day Readmissions and Reoperations in the 40–49 and 80–89 Age Groups.

**Table 1 materials-16-07012-t001:** Patient Demographics, Baseline Characteristics, and Comorbidities Stratified by Age Group.

		Age Group			
	Overall	40–49	50–79	80–89	*p*-Value
N	325,499	8895	286,988	28,603	
Patient Sex, n (%)					<0.001
Female					
Male	125,980 (38.7)	3293 (37.0)	111,622 (38.9)	10,686 (37.4)	
Patient Race, n (%)					<0.001
White	230,637 (86.6)	6105 (78.3)	203,433 (86.4)	20,415 (91.1)	
Black or African American	26,221 (9.8)	1440 (18.5)	23,378 (9.9)	1229 (5.5)	
Asian	6757 (2.5)	103 (1.3)	5991 (2.5)	643 (2.9)	
American Indian or Alaska Native	1710 (0.6)	112 (1.4)	1519 (0.6)	68 (0.3)	
Native Hawaiian or Pacific Islander	1144 (0.4)	40 (0.5)	1032 (0.4)	63 (0.3)	
Hispanic Ethnicity, n (%)	18,261 (6.8)	741 (9.4)	16,142 (6.8)	1280 (5.7)	<0.001
Outpatient, n (%)	48,463 (14.9)	1228 (13.8)	43,711 (15.2)	3350 (11.7)	<0.001
BMI, mean (sd)	33.03 (6.63)	36.60 (7.62)	33.30 (6.58)	29.17 (5.09)	<0.001
ASA Class, n (%)					<0.001
1-No Disturb	5552 (1.7)	336 (3.8)	4971 (1.7)	180 (0.6)	
2-Mild Disturb	154,865 (47.6)	4585 (51.6)	139,198 (48.6)	10,557 (37.0)	
3-Severe Disturb	159,357 (49.0)	3863 (43.5)	138,122 (48.2)	16,964 (59.4)	
4–5 Life Threat/Moribund	5360 (1.6)	98 (1.1)	4391 (1.5)	857 (3.0)	
Anesthesia Type = General, n (%)	131,259 (40.3)	4365 (49.1)	115,854 (40.4)	10,470 (36.6)	<0.001
Wound Class, n (%)					0.002
1-Clean	324,427 (99.7)	8844 (99.4)	286,052 (99.7)	28,527 (99.7)	
2-Clean/Contaminated	833 (0.3)	39 (0.4)	731 (0.3)	57 (0.2)	
3-Contaminated	128 (0.0)	6 (0.1)	112 (0.0)	10 (0.0)	
4-Dirty/Infected	111 (0.0)	6 (0.1)	93 (0.0)	9 (0.0)	
Financial Status = Totally/Partially Dependent, n (%)	3083 (1.0)	76 (0.9)	2368 (0.8)	619 (2.2)	<0.001
Operative Year, n (%)					<0.001
2015	44,283 (13.6)	1449 (16.3)	38,842 (13.5)	3823 (13.4)	
2016	53,699 (16.5)	1622 (18.2)	47,242 (16.5)	4650 (16.3)	
2017	57,898 (17.8)	1629 (18.3)	51,192 (17.8)	4912 (17.2)	
2018	59,744 (18.4)	1564 (17.6)	52,531 (18.3)	5459 (19.1)	
2019	64,509 (19.8)	1495 (16.8)	56,791 (19.8)	6047 (21.1)	
2020	45,366 (13.9)	1136 (12.8)	40,390 (14.1)	3712 (13.0)	
Diabetes, n (%)					<0.001
No	14,111 (4.3)	288 (3.2)	12,870 (4.5)	936 (3.3)	
Insulin	265,790 (81.7)	7820 (87.9)	232,685 (81.1)	24,341 (85.1)	
Non-Insulin	45,598 (14.0)	787 (8.8)	41,433 (14.4)	3326 (11.6)	
Smoke = Yes (%)	25,652 (7.9)	1897 (21.3)	23,001 (8.0)	505 (1.8)	<0.001
Dyspnea = Yes (%)	17,687 (5.4)	312 (3.5)	15,197 (5.3)	2146 (7.5)	<0.001
COPD = Yes (%)	11,013 (3.4)	211 (2.4)	9557 (3.3)	1237 (4.3)	<0.001
Ascites = Yes (%)	38 (0.0)	1 (0.0)	37 (0.0)	0 (0.0)	0.158
CHF = Yes (%)	1159 (0.4)	11 (0.1)	936 (0.3)	212 (0.7)	<0.001
Hypertension = Yes (%)	208,695 (64.1)	3641 (40.9)	183,395 (63.9)	21,410 (74.9)	<0.001
Acute Renal Failure = Yes (%)	82 (0.0)	5 (0.1)	63 (0.0)	14 (0.0)	0.004
Dialysis = Yes (%)	538 (0.2)	22 (0.2)	471 (0.2)	43 (0.2)	0.133
Disseminated cancer = Yes (%)	404 (0.1)	11 (0.1)	328 (0.1)	61 (0.2)	<0.001
Open Wound = Yes (%)	388 (0.1)	10 (0.1)	331 (0.1)	47 (0.2)	0.072
Steroid = Yes (%)	11,482 (3.5)	471 (5.3)	10,059 (3.5)	818 (2.9)	<0.001
Weight Loss = Yes (%)	286 (0.1)	6 (0.1)	248 (0.1)	32 (0.1)	0.307
Bleeding Disorder= Yes (%)	6108 (1.9)	101 (1.1)	5054 (1.8)	927 (3.2)	<0.001
Transfusion = Yes (%)	86 (0.0)	0 (0.0)	61 (0.0)	23 (0.1)	<0.001

**Table 2 materials-16-07012-t002:** Univariable Analysis of Perioperative Outcomes and 30-Day Complications.

	Age Group				
	Overall	40–49	50–79	80–89	*p*-Value
N	325,499	8895	286,988	28,603	
Operative Time in Minutes, mean (sd)	88.92 (33.50)	97.36 (39.50)	89.13 (33.51)	83.45 (29.51)	<0.001
Total Hospital LOS in Days, mean (sd)	2.17 (2.12)	2.09 (2.45)	2.12 (2.04)	2.68 (2.65)	<0.001
Discharge Destination = Not Home, n (%)	44,328 (13.6)	621 (7.0)	34,854 (12.1)	8785 (30.7)	<0.001
Wound Disruption, n (%)	687 (0.2)	25 (0.3)	587 (0.2)	72 (0.3)	0.086
Pneumonia, n (%)	903 (0.3)	19 (0.2)	691 (0.2)	193 (0.7)	<0.001
Unplanned Intubation, n (%)	379 (0.1)	11 (0.1)	301 (0.1)	67 (0.2)	<0.001
Pulmonary Embolism, n (%)	1413 (0.4)	26 (0.3)	1189 (0.4)	198 (0.7)	<0.001
Acute Renal Failure, n (%)	144 (0.0)	1 (0.0)	116 (0.0)	27 (0.1)	<0.001
Urinary Tract Infection, n (%)	2166 (0.7)	23 (0.3)	1771 (0.6)	371 (1.3)	<0.001
Stroke/CVA, n (%)	243 (0.1)	1 (0.0)	182 (0.1)	60 (0.2)	<0.001
Cardiac Arrest Requiring CPR, n (%)	218 (0.1)	5 (0.1)	165 (0.1)	47 (0.2)	<0.001
Myocardial Infarction, n (%)	607 (0.2)	5 (0.1)	455 (0.2)	147 (0.5)	<0.001
Transfusions/Intraop/Postop, n (%)	3734 (1.1)	84 (0.9)	2905 (1.0)	723 (2.5)	<0.001
DVT Requiring Therapy, n (%)	2231 (0.7)	57 (0.6)	1878 (0.7)	289 (1.0)	<0.001
Sepsis, n (%)	523 (0.2)	13 (0.1)	445 (0.2)	64 (0.2)	0.021
Reoperation, n (%)	3444 (1.1)	127 (1.4)	2921 (1.0)	380 (1.3)	<0.001
Readmission, n (%)	9674 (3.0)	244 (2.7)	7897 (2.8)	1508 (5.3)	<0.001

**Table 3 materials-16-07012-t003:** Age Group Predictive of Outcomes in Multivariable Regression.

	Age Group, Reference Group = 50–79 Year			
	40–49		80–89	
Outcomes	OR [95% CI]	*p*-Value	OR [95% CI]	*p*-Value
Operative Time	1.08 (1.06–1.09)	<0.001	0.95 (0.94–0.96)	<0.001
Total Hospital Length of Stay	0.93 (0.89–0.97)	0.001	1.49 (1.45–1.53)	<0.001
Discharge Destination	0.41 (0.38–0.45)	<0.001	3.78 (3.65–3.91)	<0.001
Wound Disruption	0.96 (0.56–1.54)	0.884	1.26 (0.93–1.68)	0.123
Pneumonia	0.83 (0.46–1.35)	0.484	2.14 (1.76–2.59)	<0.001
Unplanned Intubation	1.09 (0.54–1.96)	0.785	2.05 (1.49–2.76)	<0.001
Pulmonary Embolism	0.7 (0.45–1.04)	0.098	1.71 (1.41–2.07)	<0.001
Acute Renal Failure	0.27 (0.02–1.21)	0.190	2.4 (1.44–3.85)	<0.001
Urinary Tract Infection	0.45 (0.28–0.68)	<0.001	1.91 (1.67–2.18)	<0.001
Stroke/CVA	0.23 (0.01–1.02)	0.142	2.73 (1.92–3.84)	<0.001
Cardiac Arrest Requiring CPR	1.1 (0.39–2.43)	0.840	2.93 (1.95–4.3)	<0.001
Myocardial Infarction	0.54 (0.19–1.18)	0.176	1.94 (1.49–2.49)	<0.001
Transfusions/Intraop/Postop	1.06 (0.83–1.32)	0.628	1.77 (1.6–1.96)	<0.001
DVT Requiring Therapy	0.92 (0.68–1.23)	0.603	1.5 (1.3–1.74)	<0.001
Sepsis	1.02 (0.54–1.74)	0.953	1.42 (1.05–1.9)	0.020
Reoperation	1.26 (1.02–1.53)	0.024	1.31 (1.16–1.49)	<0.001
Readmission	0.99 (0.86–1.14)	0.924	1.88 (1.76–2.01)	<0.001

## Data Availability

All data was taken from The National Surgical Quality Improvement Program database.

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
