# Peer review of "Higher Risk of Reoperation after Total Knee Arthroplasty in Young and Elderly Patients"

_materials, 2023, doi:10.3390/ma16217012_

Round 1

Reviewer 1 Report

Comments and Suggestions for Authors

Dear Authors, 

I read your paper with interest, and have the following comments:

Title: OK

Abstract: OK

Introduction:

"While the relationship between complications and age has been previously explored, many of the previous studies are flawed in various ways." Which ways? Please elaborate and highlight the gaps this paper will address.

Methods: OK

Results: OK

Discussion: OK

Limitations: OK

Conclusion: OK

References: OK 

Comments on the Quality of English Language

Good.

Author Response

Reviewer one:

*While the relationship between complications and age has been previously explored, many of the previous studies are flawed in various ways." Which ways? Please elaborate and highlight the gaps this paper will address.

Thank you for this comment. We have included the following text:

including single institution, non nationally-representative sampling21 and use of less recent data.13

Reviewer 2 Report

Comments and Suggestions for Authors

Thank you very much for the opportunity to review this interesting manuscript entitled "Higher risk of reoperation after total knee arthropasty in young and elderly patients". I request to the authors to answer my concerns:

1. In template, authors name and affilitations must appear.

2. In abstract, TKA abbreviations have been defined twice, in line 1 and 3.

3. In abstract, can authors provide statistical data in results?

4. Keywords: Is more adequate not to use abbreviations (NSQIP).

5. Information in introduction about TKA is very vague, in paragraph 1. Can authors increase it?. 

6. In paragraph 2, infections is a most common cause to resurgery. Can authors explain it.

7. At the end of the introduction, the objective must appear.

8. Please, be consistent in the use of TKA abbreviation. For example, in methods, line 4 appear TKA and in line 5 "total knee arthroplasty". Please, revise all the manuscript.

8. Please, define inclusion and exclusion criteria.

9. Tables do not present the journal style format.

In my opinion, the manuscript is interesting, although it is not well presented. I think that authors must be more careful with the journal style format. Please, revise the author guidelines for Materials.

Author Response

Reviewer two:

Thank vou very much for the opportunity to review this interesting

manuscript entitled "Higher risk of reoperation after total knee

arthropasty in voung and elderly patients". I request to the authors to

answer my concerns:

  • In template, authors name and affilitations must appear.

Thank you for this comment. We will check with the editorial staff regarding formatting.

  • In abstract, TKA abbreviations have been defined twice, in line 1 and 3.

Thank you for this comment. We have made this correction.

  • In abstract, can authors provide statistical data in results?

Thank you for this comment. We have included statistical data in the results.

  • Keywords: Is more adequate not to use abbreviations (NSQIP).

Thank you for this comment. We have added the abbreviation here.

  • Information in introduction about TKA is very vague, in paragraph
  1. Can authors increase it?.

Thank you for this comment. We have included more information on what total knee arthroplasty is for the lay reader.

  1. In paragraph 2, infections is a most common cause to resurgery.

Can authors explain it.

Thank you for this comment. Infection does not appear to always be the most common reason for repeat operation in our experience and literature interpretation.

  • At the end of the introduction, the objective must appear.

Thank you for this comment. We have included the objectives of the paper.

  • Please, be consistent in the use of TKA abbreviation. For example, in methods, line 4 appear TKA and in line 5 "total knee arthroplasty".

Please, revise all the manuscript.

Great catch, we have corrected throughout the manuscript.

  • Please, define inclusion and exclusion criteria.

Thank you for this comment. inclusion and exclusion criteria have been included.

  • Tables do not present the journal style format.

In my opinion, the manuscript is interesting, although it is not well presented. I think that authors must be more careful with the journal style format. Please, revise the author guidelines for Materials.

I have attempted to edit the tables and will additionally speak with the editorial staff regarding improving the tables.

Reviewer 3 Report

Comments and Suggestions for Authors

Dear Authors, 

the paper is very nicely presented. I have no substantive comments. On the other hand, I have editorial comments.

Probably, Editors will ask to improve the appearance of Tables. In the Tables there should not be such large spaces between the lines, because the table template for the journal looks different. Should there be a reference to Table 3 on page 8? On page 9, I do not understand the title of the 30th -day Complication. It is worth writing this title in words. Figure 1 should be elaborated because the gray text, the orange text and the blue text overlap, resulting in a lack of readability. From what I remember the captions for the figures should be below them. I also suggest separating Figure 1 from Table 3 in text. In addition, for tables I suggest merging the first lines, which will increase their readability.

Reviewing is difficult due to the lack of line numbering, I don't know why this is so.

Author Response

Reviewer 3:

the paper is very nicely presented. I have no substantive comments.

On the other hand, I have editorial comments.

Probably, Editors will ask to improve the appearance of Tables. In the Tables there should not be such large spaces between the lines, because the table template for the journal looks different.

I have attempted to edit the tables and will additionally speak with the editorial staff regarding improving the tables.

Should there be a reference to Table 3 on page 8?

Thank you for this comment. We have ensured that there is a reference to Table 3.

On page 9, I do not understand the title of the 30th -day Complication. It is worth writing this title in words.

Thank you for this comment. We have clarified this. 

Figure 1 should be elaborated because the gray text, the orange text and the blue text overlap, resulting in a lack of readability.

Thank you for this comment. We have edited to improve readability.

From what I remember the captions for the figures should be below them. I also suggest separating Figure 1 from Table 3 in text. In addition, for tables I suggest merging the first lines, which will increase their readability.

Thank you for this comment. We have merged the first lines as you suggest. We have separated Figure 1 from Table 3. Moreover, we will discuss with editorial staff regarding table and figure legend formatting.  

Reviewing is difficult due to the lack of line numbering, I don't know why this is so.

Thank you for this comment. We have added line numbering.

Review materials-2625103 1

I was asked to evaluate the paper titled 'Higher Risk of Reoperation after Total Knee

Arthroplasty in Young and Elderly Patients.' The study aims to analyze the impact of age on

total knee arthroplasty (TKA) outcomes using the National Surgical Quality Improvemen t

Program (NSQIP) database from 2015-2020. In my opinion, the paper is interesting and

contains relevant information on the subject matter. The manuscript presents data on patients

in various age groups following TKA, which could lead to significant citations when

discussing topics related to post-TKA functioning. This data will also facilitate a better

interpretation of potential results.

As the authors highlighted, this study emphasizes the importance of age-specific counseling

for TKA patients and provides valuable insights into the evolving landscape of knee

replacement surgery.

I have no major comments on the methodology.

Thank you for your kind comments.

In my opinion, it is worth combining the 'Data Variables' and 'Statistical Analysis' chapters .

Such a combination would enable a smoother description of the data, including details about

the tests used for analysis.

Thank you for this comment. We have combined these chapters.

Specifically, please include information on whether the normality of the distribution of analyzed parameters was checked, and if so, which test was used.

Thank you for this comment. We have included information regarding testing for normality.

Also , clarify whether the variables had normal distributions, and if they did, provide justification for using the ANOVA test mentioned in line 82.

Thank you for this comment. We have included information regarding testing for normality, and once normality was confirmed, we used ANOVA, as this is the correct test for univariate testing of multiple means that have normal distribution.

Additionally, mention the post-hoc test that was employed. This information should be included by the authors in the 'Statistical Analysis ' section (lines 79-90).

Thank you for this comment. Multivariate regression was used as post hoc testing. I have clarified this.

Round 2

Reviewer 2 Report

Comments and Suggestions for Authors

Thank you